# xVal: A Continuous Number Encoding
# for Large Language Models

**Siavash Golkar**[*,1]    **Mariel Pettee**[1,2]    **Michael Eickenberg**[†,1]    **Alberto Bietti**[†,1]

**Miles Cranmer**[3]    **Geraud Krawezik**[1]    **Francois Lanusse**[1,4]    **Michael McCabe**[1,5]

**Ruben Ohana**[1]    **Liam Parker**[1]    **Bruno Régaldo-Saint Blancard**[1]

**Tiberiu Tesileanu**[1]    **Kyunghyun Cho**[6,7,8]    **Shirley Ho**[1,6,9]

The Polymathic AI Collaboration

[1] Flatiron Institute    [2] Lawrence Berkeley National Laboratory    [3] University of Cambridge

[4] Université Paris-Saclay, Université Paris Cité, CEA, CNRS, AIM

[5] University of Colorado Boulder    [6] New York University    [7] Prescient Design, Genentech

[8] CIFAR Fellow    [9] Princeton University

## Abstract

Large Language Models have not yet been broadly adapted for the analysis of scientific datasets due in part to the unique difficulties of tokenizing numbers. We propose XVAL, a numerical encoding scheme that represents any real number using just a single token. XVAL represents a real number by scaling a dedicated embedding vector by the number value. Combined with a modified number-inference approach, this strategy renders the model end-to-end continuous when considered as a map from the numbers of the input string to those of the output string. This leads to an inductive bias that is generally more suitable for applications in scientific domains. We evaluate our proposal on a number of synthetic and real-world datasets. Compared with existing number encoding schemes, we find that XVAL is more token-efficient and demonstrates improved generalization.

## 1    Introduction

Even as Large Language Models (LLMs) exhibit sophisticated behavior in the generation and analysis of textual data, the scientific community has seen little success in applying these models to datasets consisting mostly of numerical values. LLMs have historically struggled to solve simple arithmetic problems such as multi-digit multiplication [1] and have a tendency to "confabulate" answers [2, 3].

---

[*]Contact: `siavash.golkar@gmail.com`
[†]Equal contribution.

NeurIPS 2023 AI for Science Workshop.

Standard LLM tokenization schemes do not inherently capture the precise quantitative properties that distinguish numerical data from other natural language inputs [4, 5].

Recent work has explored several potential improvements for encoding numerical information as inputs to language models (see [6] for a review). For instance, numbers can be encoded digit-by-digit, in scientific notation format, or in base-10 format. [7] maps numbers onto a finite set of "prototype numerals", while [8] enforces constraints such that the cosine distances between the embeddings of numbers reflects their actual mathematical distance. Transformers that use such encodings have been shown to successfully solve various mathematical problems, such as linear algebra problems including matrix multiplication [9].

Despite these improvements, many challenges remain unresolved. Language models are known to exploit shortcuts and spurious correlations in the data [10, 11, 1] and still struggle with interpolation and out-of-distribution generalization in mathematical problems and in scientific domains [12, 13]. Functions appearing in such domains are often continuous or smooth, with certain exceptions such as points of criticality. Similarly, transformer architectures applied to vision and audio domains [e.g., 14, 15] typically treat numbers continuously without tokenization [see however 16, 17], but these models typically require highly structured inputs, and cannot be applied to datasets with arbitrary sequences of text and numbers. On the other hand, when encoding numbers as text, LLMs are inherently discontinuous in both the encoding and decoding stages. While discrete models can (and do) learn to approximate continuous functions [18], this can be more challenging and less sample efficient compared to models that have continuity built-in by construction, as in many non-parametric regression models [19]. In order to overcome this inherent challenge, it is necessary to impose the appropriate inductive bias based on our knowledge of the continuous nature of numbers.

We introduce XVAL, an inherently continuous method of encoding numerical values in Large Language Models. By encoding the magnitude of numerical values multiplicatively and orienting them in a learnable direction within the embedding space, XVAL substantially changes how numbers are processed and interpreted by transformer architectures. This leads to an encoding scheme with a single vocabulary element that also encodes every number as a single token. XVAL is therefore both token-efficient and has minimal vocabulary footprint.

Coupled with a modified number-inference paradigm, XVAL allows a transformer model to be continuous (or smooth given smooth non-linearities) when considered as a map between the numbers of the input string and those of the output. We expect that this leads to a better inductive bias when the functions being approximated are continuous or smooth. We evaluate XVAL on a number of synthetic and real-world scientific datasets and compare with existing number encoding schemes. We demonstrate that XVAL is both more token-efficient and exhibits better interpolation properties.

**Our contributions**

- We introduce XVAL, a novel approach for encoding numerical values in Large Language models. Compared to existing number encoding schemes, XVAL is both token-efficient and has a minimal vocabulary footprint.

- We introduce a modified number inference scheme that, in conjunction with XVAL, renders transformer models continuous as a function of the numerical values appearing in the text.

- We evaluate XVAL and a number of existing number encoding schemes on several synthetic and real world datasets. We demonstrate that XVAL consistently provides better interpolation properties and is more compute-efficient than prior work.

## 2   Methods

**xVal: A Continuous Number Encoding.**   Instead of using different tokens for different digits or composite numbers, XVAL embeds numerical values directly along a specific learnable direction of the embedding space. A diagram of this procedure can be seen in Fig. 1. Specifically, given a string input $x$ comprising both numbers and text, we first parse $x$ to extract all the numerical values and collect them in a separate list $x_{\text{num}}$. We then construct a new string $x_{\text{text}}$ by replacing all numbers in $x$ with a designated token [NUM] that acts as a placeholder for numerical values. We tokenize and embed $x_{\text{text}}$, arriving at $h_{\text{text}}$. We then multiply the embedding of each appearance of the [NUM] token with its associated numerical value in $x_{\text{num}}$. This process can be done efficiently by defining a new

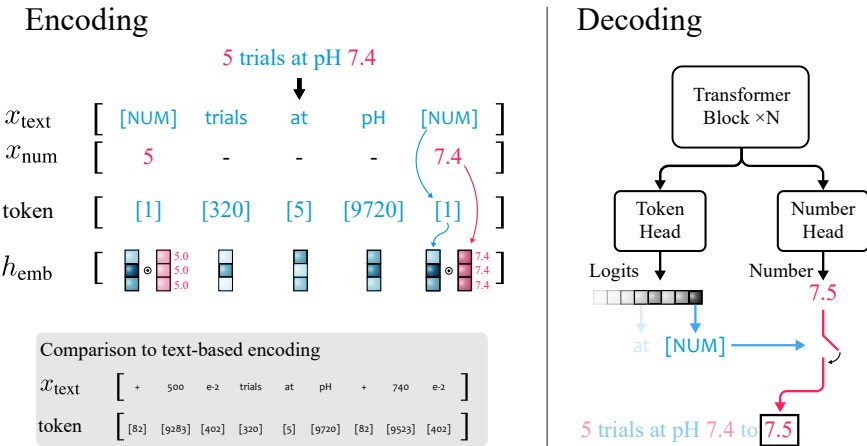

Figure 1: (Left) XVAL is contrasted with the P1000 text-based numerical encoding scheme. (Right) we illustrate how numbers are addressed within the decoder.

list $h_{\text{num}}$ by scattering $x_{\text{num}}$ to have the same length as the tokenized $x_{\text{text}}$ and inserting a 1 for any token other than [NUM]. The final embedding of the sample is $h_{\text{emb}} = h_{\text{num}} \times h_{\text{text}}$, which is then fed to the transformer trunk.

This encoding process can be performed both for masked language modeling (MLM) and autoregressive (AR) generation. During training, in cases where MLM is used, we simultaneously mask both $h_{\text{text}}$ and $h_{\text{num}}$, i.e., if the token being masked is a [NUM] token, we replace the corresponding number in $h_{\text{num}}$ with 1.

**Implicit normalization via layer-norm.** In our implementation, the multiplicative embedding of XVAL is followed by the addition of a positional encoding vector and then a layer-norm in the first transformer block. The effect of the layer-norm is to normalize the embedding of each token on a per-sample basis. When the added positional embeddings are not collinear to the embedding of the [NUM] token, the scalar value is effectively passed through a non-linear rescaling function. Indeed, denoting $u \in \mathbb{R}^d$ as the embedding of [NUM], $p \in \mathbb{R}^d$ as the positional embedding, and $x \in \mathbb{R}$ as the scalar to be encoded, and assuming for simplicity $u \cdot p = 0$ with $\|u\| = \|p\| = 1$, we have

$$u \cdot \frac{xu + p}{\|xu + p\|} = \frac{x}{\sqrt{1 + x^2}}, \tag{1}$$

such that the value $x$ is still encoded in the same direction $u$. Figure 9 shows that such a property approximately holds empirically up to a constant after training, and we found these curves to be near-identical for any positional embedding.

This normalization property implies that the dynamic range of XVAL is more limited than those of other text-based encoding schemes. In the experiments of this paper, we normalize numbers in the text corpus such that they fall within the range $[-5, 5]$ as a preprocessing step before training.

**Numerical value inference.** XVAL defines an embedding that is continuous in the numerical values of the input. However, if we use a multi-class classification task as our output and training algorithm, the model as a whole will not be end-to-end continuous when considering the map from the input numbers to the output numbers. For this reason, we treat numbers separately at the output layer. This process is illustrated in the right-hand portion of Fig. 1.

As is standard practice in transformer-based language models, we define a token head that outputs a probability distribution of the tokens of the vocabulary. However, since our formalism replaces numbers with the [NUM] token, this head does not carry any information about the number value. We therefore introduce a new number head with a scalar output, trained via mean squared error (MSE) loss, to recover the numerical value associated with each instance of the [NUM] token. For any input, we first look at the output of the token head. If the generated token is the [NUM] token, we then look at the number head to fill in the value for this token. As shown in Section 3, since the transformer is

now end-to-end continuous when inferring numerical values, it performs better when interpolating to previously unseen values.

## 3 Experiments

In this section, we evaluate the performance of XVAL and highlight its strengths and weaknesses compared to existing numerical encoding algorithms. In particular, we look at three datasets: a synthetic dataset of arithmetic operations, a dataset of global temperature data, and a dataset of planetary orbit simulations.

For our transformer models, we use an architecture based on GPT-2 [20]. Details of our specific architecture are included in Appendix A). We explore the effects of various architectural design choices in Appendix B.5.

Table 1: Comparison of XVAL with four other number encodings. XVAL is more token-efficient and has a minimal vocabulary footprint. Vocabulary size differs from [9] because we only consider exponents from 1E–8 to 1E+8.

| Encoding | Tokens | $-6.02 \times 10^1$ | Tokens per number | Vocabulary Size |
|---|---|---|---|---|
| P10 | {±, d, E±d} | [-, 6, 0, 2, E-1] | 5 | 28 |
| P1000 | {±, ddd, E±d} | [-, 602, E-1] | 3 | 918 |
| B1999 | {±ddd, E±d} | [-602, E-1] | 2 | 1816 |
| FP15 | {±ddd E±d} | [-602 E-1] | 1 | 28800 |
| XVAL | {[NUM]} | [NUM] | 1 | 1 |

**Comparison with other number encodings.** We compare the performance of XVAL with four other number encodings, following the notation of [9]. In these encodings, numbers are first processed into the format ±ddd E±d. The encodings are then determined by which parts of this format are encoded as single or multiple tokens. These range from encodings with limited vocabulary size but high number of tokens per number, leading to longer encoded sequence lengths (e.g., P10), to those with very large vocabulary footprints but only one token per number, leading to shorter encoded sequence lengths (e.g., FP15). XVAL provides a minimal vocabulary footprint and uses just a single token per number, leading to the shortest sequence lengths. A summary of these encodings and an example can be seen in Table 1.

Number encodings that do not lead to a fixed number of tokens for all numbers (e.g., learned Byte Pair Encoding [21] used in GPT-2 [20]) can lead to erratic behaviors where the transformer learns spurious correlations that exist between the length of the encoded numbers in the dataset. An example of this type of behavior is shown in Appendix B.4.

**Learning Arithmetic.** Simple arithmetic problems have acted as a test bed for probing the mathematical reasoning abilities of language models [1]. In this section, we investigate the effect of the number encoding scheme on the ability of language models to perform multi-digit multiplications as well as multi-operand mathematical operations. Multi-digit multiplication is a notably challenging task for even the largest LLMs [22]. [1] show that GPT-4 achieves only 59% zero-shot accuracy on three-digit multiplication problems, while its accuracy for four- and five-digit multiplication drops to 4% and 0%, respectively.

We designed a dataset of multi-operand mathematical operations. We used random binary trees combining a fixed number of operands (2, 3, or 4) using the binary operators of addition, subtraction, and multiplication. build a dataset in which each sample is an arithmetic statement such as ((1.32 * 32.1) + (1.42-8.20)) = 35.592. We then processed the samples according to the processing requirements of each number-encoding scheme. The task is evaluation of the expression on the left-hand side of the equation, implemented as a mask completion, where the right-hand-side number is masked. Table 2 shows the adjusted $R^2$ scores results on this task. XVAL performs remarkably well on this task. For further experiments on multiplication using these encoding schemes see Sec. B.1.

Arithmetic experiments alone are not sufficient for fully evaluating the mathematical abilities of language models. The samples in these datasets are often short sequences and the underlying

Table 2: Arithmetic evaluation task of random binary trees combining different numbers of operands with addition, subtraction, and multiplication. $R^2$ measured between true expression value and transformer prediction.

| Encoding | 2 operands | 3 operands | 4 operands |
|---|---|---|---|
| P10 | 0.998 | 0.996 | 0.992 |
| P1000 | 0.991 | 0.990 | 0.991 |
| FP15 | 0.993 | 0.981 | 0.935 |
| XVAL | 0.99998 | 0.99994 | 0.99998 |

data manifold is low-dimensional. These problems therefore do not push the boundary of what is computationally possible with LLMs.

**Temperature forecasting.** As an example of real-world scientific analysis, we look at the task of temperature forecasting. In this experiment, we construct a dataset as a subset of the ERA5 global climate dataset [23]. For simplicity, we only focus on the surface temperature data (T2m field in ERA5). We split the dataset into individual samples, where each sample includes 2–4 days of surface temperature data (normalized to have unit variance) as well as the latitude and longitude from 60–90 randomly selected reporting stations. We also include the time of the first included timestep. We encode the coordinates by using the sine of the latitude and the sine and cosine of the longitude such that we preserve the periodicity. Similarly, we encode the time of year and time of day using the sine and cosine of the position along the 24 hour and 365 day cycles. We include all this information in a JSON format as follows[3]:

```
{'description':{'coords':[[1,-.32,.95] ... [.96,.61,.79]],
'start':[0,1,-.026,-1]}, 'data':[[-2.6,-2.6 ... -3.2,-3.1,-3]]}
```

The `coords`, `start`, and `data` correspond to the reporting station coordinates, the time of the first sample, and the normalized temperature data, each reported separately per station and then concatenated in the data list. In this way, the model needs to parse both the textual aspects of the sample (e.g., where the commas appear to separate different parts of the data) as well as the numerical values. Furthermore, as is often the case with JSON-formatted data, the data does not have a causal format. We therefore train the language models using an MLM approach instead of the more common AR approach. We evaluate the performance of the different numerical encodings on the task of

Table 3: Performance (in MSE) and runtime of the different encodings on predicting the temperature for the next time step. "Equal Samples" columns refer to all models being trained for 500k iterations. Training was performed on 4 Nvidia H100 GPUs using Pytorch Distributed Data Parallelism.

| Method | Equal Samples | | Equal Tokens | | Equal Runtime | |
|---|---|---|---|---|---|---|
| | Loss | Runtime | Loss | Runtime | Loss | Runtime |
| P10 | 73 | 2d 22h | 73 | 2d 22h | 73 | 2d 22h |
| P1000 | 20 | 2d 2h | 23 | 3d 10h | 21 | 2d 22h |
| B1999 | 20 | 20h | 19 | 2d 23h | 19 | 2d 22h |
| FP15 | 2.14 | 19h | 1.76 | 3d 12h | 1.85 | 2d 22h |
| XVAL | **1.75** | **9h** | **1.62** | **1d 15h** | **1.51** | 2d 22h |

predicting the next temperature timestep for all reporting stations simultaneously in a held out test set. We do so by masking the tokens (and numbers, if applicable) of all the data associated with the final timestep. Because the temperature data is provided separately per station, the masks are scattered throughout the input data and are not all simply at the end of the sample.

Table 3 shows the results of this experiment. XVAL provides the best performance while taking considerably less compute time.

---

[3]For demonstration purposes, we show a few digits per number, but for both scientific datasets, all numbers are floating point numbers. For the text-based encodings, this text string is then processed according to the procedure described above.

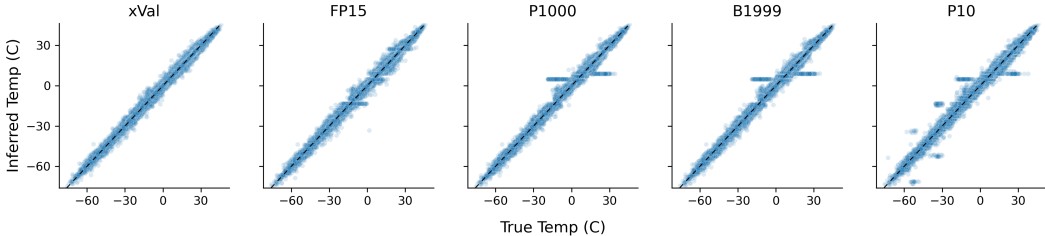

Figure 2: Performance of the encoding schemes predicting the temperature of the next timestep.

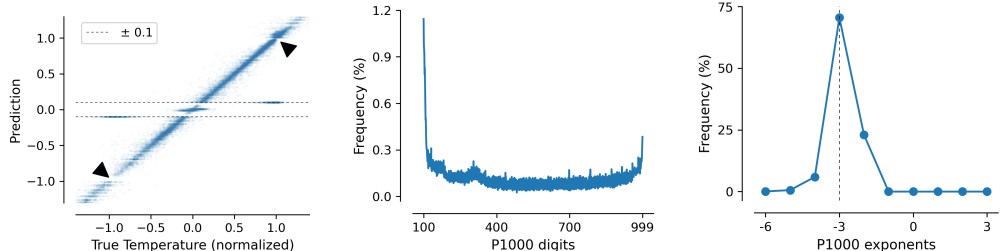

Figure 3: A failure mode of text based encoding scheme (left). Because of the distribution of the numbers in the training set (center and right), numbers that are close to ±1 (denoted by the black arrows) get misclassified as 100E-3, i.e. 0.1, the combination of the most common digit and the most common exponent in the dataset.

This task exemplifies one of the shortcomings of text-based encoding schemes: they can take advantage of spurious correlations in the data. In this case, P10, P1000 and B1999 have a tendency to predict normalized temperature ±0.1, which manifest as extended protrusions in Fig. 2. This is due to the over-abundance of this number in the dataset compared to other numbers, as seen in Fig 3. While individually, 100 and E-3 are the most common numbers and exponents in the dataset, when combined, 100E-2 is much more frequent than 100E-3. This explains why FP15, which encodes the digits and exponents as one token, does not get confused in this case. It also implies that the model has failed to learn the correct joint distribution of the numbers. In these cases, because of the tokenization scheme, the length of the tokenized samples are very long, averaging around 8000 and 5000 tokens respectively for P1000 and P10 (compared to 1800 tokens for FP15 and xVAL). The poor performance in these models might therefore be due to the the challenges of modelling long-range interactions [24].

For more details on the performance of the different encodings, as well as comparison with some non-transformer baselines, see Appendix B.2. In Appendix B.4 we look at the performance of a BPE tokenizer on this task and demonstrate how LLMs can exploit the tokenized length of the number.

**Predicting planetary orbits.** We then compare the performance of the various number encoding schemes on a simulated dataset of planetary orbits. We construct a dataset consisting of planetary motion simulations generated by the REBOUND N-body codebase [25] and integrated using IAS15 [26]. The dataset consists of 1.25 million samples, split into 80%, 10%, 10% for training, validation, and test. Each sample consists of simulation parameters (mass and orbit properties of each planet and the simulation timestep size) as well as a sequence of $(x, y)$ positions for each planet, organized in a JSON format. The details of the simulation are provided in Appendix B.3. A typical sample in this dataset is given by:

```
{'description':{'planet0':{'m':2.38, 'a':2.96, 'e':1.73},
'planet1':{'m':1.35, 'a':2.96, 'e':1.73}, ... , 'stepsize':0.2},
'data':[[[2.60,-0.75],[0.81, 0.42]],[[2.63,-0.63],[0.70,0.60]]...]}
```

We pretrain the models using MLM and evaluate the models on the task of inferring the simulation parameters, specifically the simulation timestep $\Delta t$, and the semi-major axis, eccentricity and mass of the first planet $(a_1, e_1, m_1)$ by masking the appropriate locations. The quantities $\Delta t$ and $a_1$ in

the training corpus take values that are either discrete or are sampled from intervals with gaps. This property makes these quantities a good testing ground for interpolation generalization.

Table 4: Performance of the different encodings on the planetary motion inference problem. Here, OoD implies evaluation on samples where the quantity was not seen in the training corpus. The percentages in brackets denote the fraction of the predictions that could not be parsed as numbers. When not specified, this fraction was less than 0.01%. (†) The poor performance here is because of a number of outliers that are being mis-classified.

| Method | $a_1$ | $a_1$ (OoD) | $e_1$ | $\Delta t$ | $\Delta t$ (OoD) | $m_1$ |
|---|---|---|---|---|---|---|
| P10 | $7.6 \times 10^{-4}$ | 0.0076 (1%) | 0.20 | **0.0** | 0.0036 | 1.5 |
| P1000 | $4.5 \times 10^{-6}$ | 0.048 | 0.0067 | **0.0** | 0.011 | 0.74 |
| B1999 | $\mathbf{3.6 \times 10^{-6}}$ | 0.11 | 0.0057 | **0.0** | 0.022 | 0.44 |
| FP15 | $4.0 \times 10^{-6}$ | 0.050 | $\mathbf{3.6 \times 10^{-4}}$ | $0.0065^{\dagger}$ | 0.0075 (0.2%) | **0.37** |
| xVAL | $6.4 \times 10^{-5}$ | **0.0010** | 0.0020 | $6.6 \times 10^{-5}$ | **0.0021** | 1.4 |

The results of this test are presented in Table 4. In the numerical encoding schemes other than xVAL, we see an overall inverse relationship between performance in- and out-of-distribution. For example, P10—the encoding with the fewest vocabulary elements—provides the worst in-distribution performance but is best on out of distribution tasks. This is an example of the bias/variance trade-off applied to the number of vocabulary elements.

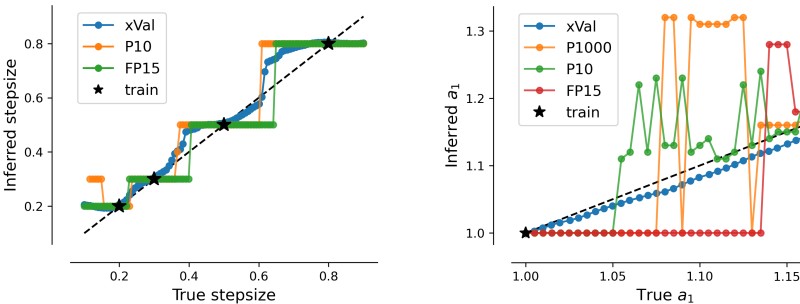

Figure 4: Out of distribution generalization properties of the different number encoding schemes. Left: Inferring $\Delta t$, which takes discrete values in the training set. Right: Inferring $a_1$ which is either 1 or > 1.16 in the training set. Because of the generation procedure, taking $a_1 \rightarrow 1.16$ here does not result in an in-train-distribution sample.

In comparison, we see that xVAL provides the best out-of-distribution performance while staying competitive in-distribution (with one exception). The out-of-distribution performance of these encoding methods can be seen in Fig. 4. Here we see that the text-based encodings, with the exception of P10, simply do not predict any number that they did not explicitly see for this parameter in the training corpus. As expected from a function that is continuous by construction, xVAL continuously interpolates between the values seen in the training set and offers much better performance.

Figure 4 shows that the predictions coming from the text-based encodings can be discontinuous when evaluated out-of-distribution. This discontinuity has two potential sources: the discontinuous nature of the number embeddings and the argmax that is taken over the logits during inference. Since the encodings of the number tokens in text-based encodings have been shown to form continuous-looking structures (see Sec. B.6 and [27, 18]), it is possible that the discontinuiuty is only a side effect of the argmax and that the logits themselves vary more smoothly. Figure 5 shows an example of the logits of the P1000 encoding when predicting the step-size out-of-distribution. Here, the color lines denote the highest-value logits, with the other logits carrying negligible weight. The dashed gray lines denote the values of the step-size seen in the training set. We see that these lines are smooth in neither small or larger scales. We expect that this is a combination of the text-based number encodings' discrete embedding schemes together with the cross-entropy training paradigm that does not incorporate number distances into the loss.

**Results summary.**    It is evident that embedding the magnitude of numbers directly, as in XVAL, leads to a different inductive bias than treating numbers as tokenized text. This can be clearly seen in the varying performance of these language models in different tasks. When predicting the next timestep in the temperature dataset, XVAL provides by far the best results. On the other hand, in the mass prediction task, it fails to learn the correct relationship, along with vocabulary-sparse P10.

Where XVAL excels is in out-of-distribution performance, while the text-based encoding schemes fail to interpolate properly. The best interpolation for the text-based encodings is given by the vocabulary-sparse P10, which performs poorly on the in-distribution tasks. However, it often performs poorly when

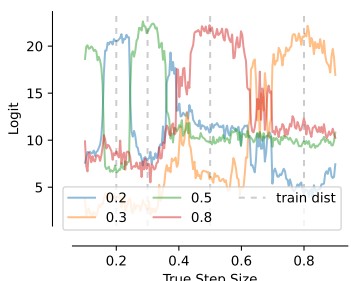

Figure 5: An example of the logits of model trained with P1000 encoding.

evaluated on in-distribution tasks. The the extra encoding length of P10 also makes it prohibitively expensive to deploy as can be seen in Table 3. On the other hand, FP15 provides the best in-distribution performance but it has poor interpolation properties and expensive embedding cost. Overall, XVAL provides the best mix of in-distribution and out-of-distribution performance. Moreover, it is the most computationally efficient of the encoding schemes we considered.

**Failure modes.**    There are a number of ways that number inference via a large language model can fail. The language model can predict a non-numeric token in the place of the number, leading to an invalid prediction. These are denoted in the percentages in brackets in Table 4, shown only when the percentage exceeded 0.01%. This failure mode is uncommon and becomes less frequent the more the model is trained. Another failure mode is when the model exploits spurious correlations. For example, the model can learn the distribution of the digits, as discussed in the example of temperature dataset, or the length of the encoding (see Appendix B.4).

A model can also fail to learn the correct distribution. In the planetary orbits example, learning the mass of the planet is the most challenging task – all encodings struggle with this. In this task, XVAL performs uncharacteristically poorly. We suspect that this is due to the high uncertainty in estimating the mass and that a multi-modal distribution such as the categorical distribution learned by traditional LLMs would perform better. This can be seen in Fig. 6, where the predictions of P10 and XVAL are shown. While both of these models perform poorly when considering the MSE of the prediction, the multi-modal prediction of P10 would be a better starting point for capturing an uncertain distribution. We therefore suspect that generalizing the number-head such that instead of predicting a scalar for each number, it fits a mixture of Gaussians, would improve this performance. We leave explorations in this direction for future investigation.

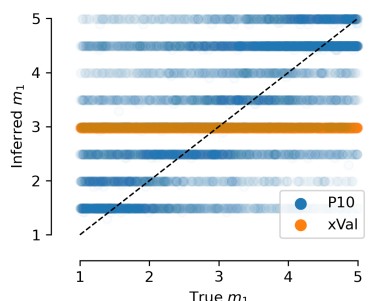

Figure 6: Different failure modes of XVAL and text-based encodings.

# 4   Discussion

In this work, we introduced XVAL, a continuous number encoding that makes transformer-based models end-to-end continuous when considered as a function mapping the numerical values of the input to those of the output. We demonstrated that even though XVAL is more token-efficient and has a minimal vocabulary footprint, it excels in numerical tasks and leads to superior performance, especially when evaluated on out-of-distribution samples. Because of the fundamentally different treatment of numbers across these cases, XVAL and text-based encodings lead to different inductive biases, making the choice of the best encoding method on a given dataset highly dependent on the problem under consideration.

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

## A  Architecture details

In our experiments, all the language models, regardless of encoding, adopt the main features of GPT-2 [20]. That is, we use absolute position encoding and the transformer blocks have layer norms prior to the attention module and the MLP (i.e., after each residual connection). We also set the width of the MLP hidden layer equal to 4 times the width of the embedding. We deviate from GPT-2 in that we initialize all the weights of the transformer blocks with a normal distribution with standard deviation given by $(2 \times \text{fan-in} \times \text{num-layers})^{-1/2}$. The dependence of the standard deviation on the number of transformer blocks is to counteract the effect of having a series of residual connections. We also do not use any biases in the trunk of the transformer. As is standard, after the transformer blocks, we have a token-head, comprised of a single linear layer, which maps the latent embedding of each token into a distribution over the vocabulary. As in GPT-2, we tie this weight to that of the embedding matrix which maps the tokens of the input to the embedding space.

For the LLMs using XVAL encoding and an MSE number-head in addition to the token head, we promote both heads (number and token) to be MLPs with one hidden layer of width equal to the embedding dimension. This was to allow the two different prediction types (the number and the distribution over the vocabulary) to be processed separately before the final prediction. In particular, we explore the possibility of having biases for the number-head and not in the token-head in Sec. B.5.

For all of our training runs, we use a cosine learning-rate schedule with warm-up. The scheduler is adjusted such that it reaches the minimum learning rate at the end of the training run.

## B  Further experimental details

### B.1  Multiplication

Table 5 reports the $R^2$ scores for multi-digit multiplication problems on several language models designed to handle numerical values. All number encodings generally perform well on this task. However, we find that some encoding schemes (P10 and FP15) show a tendency to yield a small percentage of highly erroneous predictions in some contexts, thereby reducing the $R^2$ score, while XVAL does not produce such outliers.

Table 5: Adjusted $R^2$ scores calculated between predictions and true values for the different encodings on various arithmetic datasets. (Higher is better; $R^2 = 1$ is the theoretical maximum.)

| Encoding | 3-digit Multiplication | 4-digit Multiplication | 5-digit Multiplication |
|---|---|---|---|
| P10 | 0.9989 | 0.6071 | 0.9439 |
| P1000 | 0.9997 | 0.9783 | 0.9991 |
| B1999 | 0.9998 | 0.9984 | 0.9997 |
| FP15 | 0.7119 | 0.9959 | 0.9980 |
| XVAL | 0.9986 | 0.9975 | 0.9958 |

### B.2  Temperature forecasting

#### B.2.1  Experiment details

**Dataset details.** The ERA5 dataset [23] is a high-resolution, state-of-the-art global atmospheric reanalysis product provided by the European Centre for Medium-Range Weather Forecasts (ECMWF). It is the fifth generation of ECMWF atmospheric reanalyses and represents the latest advancement in the ERA (ECMWF Re-Analysis) project. The dataset covers the period from 1979 to near-real-time and is updated regularly.

In our experiment, we take only the surface temperature of the dataset (field T2m) sampled at 8 hour intervals. For each sample, we randomly choose 60–90 of the ~1-million spatial grid points of the dataset, and include 8–16 temperature time points at 8-hour intervals (corresponding to 2–4 days), starting from a random time. We generate 1.25 million examples in this way and split it into 1 million train, 125 thousand validation and test set samples.

```
{'description':{'coords':[[1,-.32,.95] ... [.96,.61,.79]],
'start':[0,1,-.026,-1]}, 'data':[[-2.6,-2.6 ... -3.2,-3.1,-3]]}
```

The samples are individually preprocessed such that the temperature range across all samples has mean zero and standard deviation equal to 1. We also include the lattitude longitude information. To respect the periodicity of this information, we provide the sine of the lattitude and the sine and cosine of the longitude. Furthermore, we specify the starting time for each sample as the day of year and time of day. Again to respect the periodicity of these quantities, we provide the sine and cosine of the phase of these quantities.

**Architecture design hyperparameters.** For all experiments done with this dataset, we use transformers with 6 transformer blocks, each with 6 heads and each head having width 128, resulting in a embedding width of 768 (43.5M parameters).

**Training hyperparameters.** For the equal samples training runs, we train each model for 500k iterations with batch size equal to 64 samples. For the equal tokens runs, we increase the number of iterations proportionately such that the total number of tokens seen is equal. This implies: 500k samples for P10, 820k for P1000, 1.2M for B1999, and 2.3M for FP15 and XVAL. Since there is non-numeric data in the samples, the ratio of the length of the equal tokens is slightly different from the ratio of the length of each encoding scheme's tokenization length for numbers. The other hyperparameters in this task are given in Table 6.

Table 6: Training hyperparameters for the different encodings on the Temperature Forecast dataset.

| Encoding | Learning Rate | Minimum LR | Warmup | Max Context Length |
|----------|---------------|------------|--------|---------------------|
| P10 | $3 \times 10^{-5}$ | $3 \times 10^{-6}$ | 2000 | 8222 |
| B1999 | $10^{-4}$ | $10^{-5}$ | 2000 | 1251 |
| P1000 | $10^{-4}$ | $10^{-5}$ | 2000 | 5010 |
| FP15 | $10^{-4}$ | $10^{-5}$ | 2000 | 1798 |
| XVAL | $2 \times 10^{-4}$ | $2 \times 10^{-5}$ | 2000 | 1798 |

### B.2.2 Non-transformer baselines

To understand this task better, we trained a number of non-transformer baselines for comparison. These models are reported just for comparison and by no means represent the best possible non-transformer based baslines.

First, we looked at the performance of an MLP model when trained in a supervised way to predict the next time step (All stations). To deal with the varying number of locations and varying number of time-steps, we simply keep the number of locations/time-steps that is the minimum across all samples (60 locations and 8 time-steps.) We then looked at the possibility of temperature forecast based on a single reporting station (Single Station). And then on this single-station dataset, we looked at the performance on the temperature data alone (Single Station - temp all), temperature data + station coordinate (Single Station - temp + coord), and temperature data + first time step time of year (Single Station - temp + ToY).

The MLPs acting on single stations have 3 hidden layers of width 256. The MLP looking at 60 stations simultaneously is larger to validate that the poorer performance is not because of limited network size. We tried width from 256–8192 and up to 5 layers and the results remain similar.

Table 7: Temperature forecast MLP baselines

| Method | MSE Loss (C) |
|--------|--------------|
| All Stations | 2.31 |
| Single Station | 1.57 |
| Single Station - Temp only | 1.79 |
| Single Station - Temp + Coord | 1.65 |
| Single Station - Temp + ToY | 1.74 |

The results of these tests can be seen in Table 7. We see that for good performance, it is important for the model to have access to both the time of year as well as the coordinate of the reporting station. However, providing the information for multiple reporting stations at once makes the performance worse.

This implies that for the transformer model to be able to predict the temperature with MSE less than 1.7, it needs to properly parse all this information that is scattered across the different parts of the input string. XVAL was the only model to achieve MSE below that of the MLP model (Table 3) meaning that it has likely learned to leverage the temperature of other reporting stations as well.

### B.3  Planetary motion

**Dataset details.**    In this dataset we use the REBOUND N-body simulation codebase [25] and IAS15 integrator [26] to generate a number of planetary systems (with a central mass $m_\odot \equiv 1$) and follow their orbits for a number of time points. Each planetary property is drawn from a uniform prior: the number of planets $n \in [2, 4]$, mass $m/m_\odot \in [10^{-5}, 5 \cdot 10^{-5}]$, semimajor axis equally spaced for the planets between 1 and $a_f \in [1.5, 3]$ (i.e. if 3 planets and $a_f = 1.8$ then $a_1 = 1$, $a_2 = 1.4$ and $a_3 = 1.8$), eccentricity $e \in [0, 0.1]$, and starting angle in the $(x, y)$ plane equal to zero for $30\%$ of the samples and uniform $\theta \in [-\pi/6, \pi/6]$ for the remainder. These choices are made such that when generating the large number of samples required for training, we do not come across instabilities or collisions. Finally, we use an integration step-size sampled uniformly from $\{0.2, 0.3, 0.5, 0.8\}$.

We generate 1.25 million examples in this way and split it into 1 million train, 125 thousand validation and test set samples. We normalize the masses such that they take value between 1 and 5 and the eccentricities such that they are between 0 and 2. We then construct a JSON format sample including all of this information. A generic sample is given in this example.

```
{'description':{'planet0':{'m':2.38, 'a':2.96, 'e':1.73},
'planet1':{'m':1.35, 'a':2.96, 'e':1.73}, ... , 'stepsize':0.2},
'data':[[[2.60,-0.75],[0.81, 0.42]],[[2.63,-0.63],[0.70,0.60]]...]}
```

**Architecture design hyperparameters.**    Similar to the Temperature Forecasting dataset, for all experiments, we use transformers with 6 transformer blocks, each with 6 heads and each head having width 128, resulting in a embedding width of 768 (43.5M parameters).

**Training hyperparameters.**    We train each model for 500k iterations with batch size equal to 64 samples. The hyperparameters in this task are given in Table 8.

Table 8: Training hyperparameters for the different encodings on the Temperature Forecast dataset.

| Encoding | Learning Rate | Minimum LR | Warmup | Max Context Length |
|---|---|---|---|---|
| P10 | $10^{-4}$ | $10^{-5}$ | 2000 | 2707 |
| B1999 | $10^{-4}$ | $10^{-5}$ | 2000 | 1251 |
| P1000 | $10^{-4}$ | $10^{-5}$ | 2000 | 1736 |
| FP15 | $10^{-4}$ | $10^{-5}$ | 2000 | 767 |
| XVAL | $2 \times 10^{-5}$ | $2 \times 10^{-6}$ | 2000 | 767 |

### B.4  Erratic behavior of number encodings of unfixed length

In many JSON formatted datasets, the data does not follow a causal pattern, i.e. earlier entries might depend logically on latter entries. This is also the case for our JSON formatted samples. Because of this we used Masked Language Modeling (MLM) for pretraining our models. In the context of MLM, number encodings that lead to encoding lengths that vary based on the number can prove troublesome both during training and during testing. During train time, the length of the encoding acts as a cue to help the model figure what the number is. This is an example of spurious correlations that LLMs are known to exploit [10, 11, 1]. Similarly at test time, the length of the mask can bias the model toward predicting one number or another.

As a demonstration of this feature, we first preprocessed the Temperature Forecast dataset such that every number has only two significant figures and drop leading zeros for efficiency (e.g. $0.12 \rightarrow .12$).[4] We then used a tokenizer that included single and double digits as well as $\pm$, the decimal point and exponents ranging from (E-8 to E+2). In this dataset, Positive and negative floats with magnitude between 0.1 and 1 (e.g. .23 and -.34) would have encoding lengths equal to 2 and 3 and Positive and negative floats with magnitude between 0.01 and 0.1 (e.g. -.034 = 3.4E-2) would have encoding lengths 4 and 5. There are exceptions however. For example in this scheme 0.030=3E-2 has encoding length 2.

The results of this experiment can be seen in Fig. 7. We see that even though the model's overall performance is not great, it can tell with very high accuracy the numbers sign, whether or not it has absolute value greater/less than 1, or greater/less than 0.1. This is due to the fact that the model is exploiting the correlation of the numbers with the length of the encoding. We verify this by highlighting in orange the cases where in the range between 0.01 and 0.1, the number has encoding length 2, that is it does not follow the general trend mentioned above. We see that the model believes that these numbers are greater than 0.1 (which as we saw generally had encoding length 2).

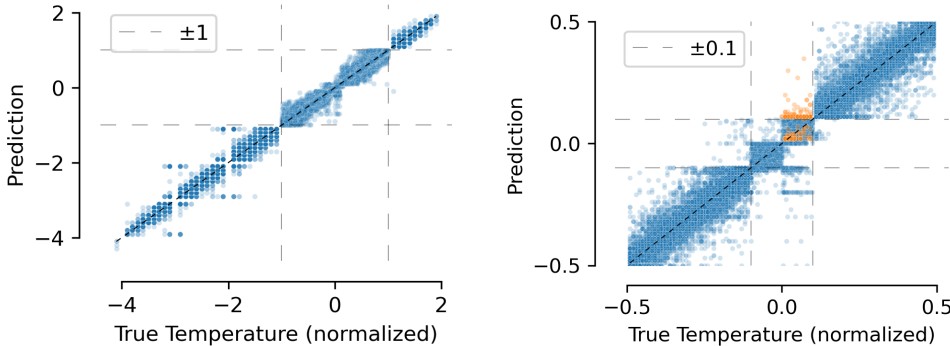

Figure 7: LLMs can exploit spurious correlations in the data. In this case, the model has learned the correlation between the number signs/values with the length of the encoding. Highlighted in orange are numbers between 0 and 0.1 that do not have the encoding length equal to 2..

## B.5 Architectural explorations

There are a number of engineering choices that we made regarding the architecture and hyperparameters of the transformer models trained with XVAL and the number head. Here, we explore the effect of these on the Temperature Forecast task. Because of the large exploration space and the high amount of compute required, we do the ablation tests on a shorter run, 100k iterations compared to 500k iterations of the main text. For this exploration, we first run all of the configurations with 4 different learning rates (2.5E-5, 5E-5, 1E-4, 2E-4). We then choose the best performing learning rate for each configuration and then run each configuration two more times with this learning rate. The result of this exploration is given in Table 9.

We summarize the various configurations that we run this experiments in and their effects as follows:

- Ratio of the final learning rate of the cosine scheduler to the initial learning rate (min-LR/LR). We found decreasing this ratio from 0.1 to 0.01 does not affect performance in this experiment. But we found that it does increase stability in longer runs.

- Turning off the layer norm prior to the MLP of the first transformer block (First Layer Norm = False). This change does not affect average performance. This is not surprising since the effect of the layer norm at this stage is simply to normalize the numbers and the numbers in this dataset are in the regime where the normalization discussed in Sec. 2 is linear.

---

[4] In the experiments of the Sec. 3, the numbers have three significant figures. Therefore the results of this section are not directly comparable to those of the main text.

Table 9: Ablation tests for the various design choices. Here Normal refers to min-LR/lr=0.1, Weight decay = 0.1 and MLM probability = 0.2, and the opposite dichotomy for the other choices.

| Configuration | Best Validation Loss | Learning Rate |
|---|---|---|
| Normal | $(6.8 \pm 0.2) \times 10^{-3}$ | 0.0002 |
| min-LR/LR = 0.01 | $(7.0 \pm 0.1) \times 10^{-3}$ | 0.0002 |
| First Layer Norm = False | $(6.8 \pm 0.5) \times 10^{-3}$ | 0.0002 |
| MLP Layer Norm = False | $(9.0 \pm 0.1) \times 10^{-3}$ | 0.0001 |
| MLM probability = 0.1 | $(8.2 \pm 0.6) \times 10^{-3}$ | 0.0002 |
| MLM probability = 0.3 | $(6.4 \pm 0.4) \times 10^{-3}$ | 0.0002 |
| Weight decay = 0.0001 | $(8.2 \pm 0.6) \times 10^{-3}$ | 0.0002 |
| Weight decay = 1 | $(5.3 \pm 0.3) \times 10^{-3}$ | 0.0002 |
| Trunk bias = True | $(6.2 \pm 0.4) \times 10^{-3}$ | 0.0002 |
| Num-head bias = False | $(6.9 \pm 0.1) \times 10^{-3}$ | 0.0002 |

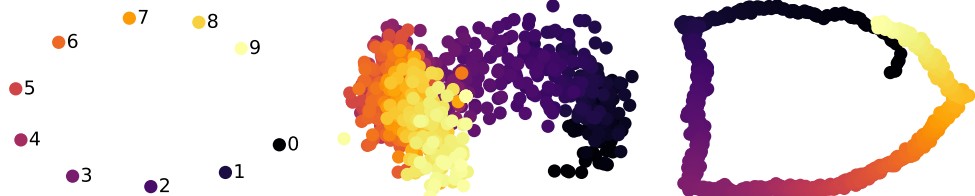

Figure 8: Two-dimensional PCA projection of the learned embeddings for mantissa tokens. (left) P10 encoding trained on the planet dataset; (center) P1000 encoding trained on the planet dataset; (right) P1000 encoding trained on the arithmetic dataset. Brighter colors denote higher number values.

- Turning off the layer norm prior to the MLPs of all transformer blocks (MLP Layer Norm = False) This change had a significant negative impact on the performance of the model.

- Changing the masking probability to 10% or 30% (default is 20%). Decreasing (resp. increasing) this probability lead to performance deterioration (resp. improvement) in this experiment. However, this seems to be dependent on the dataset as in other instances 30% seems to be too high for effective learning.

- Changing the weight decay to 0.0001 or 1 (default is 0.1). Increasing this value lead to the largest improvement. However, similar to the masking probability, this seems to be dataset dependent. The effect of increased weight decay can also depend on the length of the run.

- Including a bias in the modules of the transformer block (they are absent by default). Including this bias improved performance at the cost of increased variability.

- Turning off the bias in the number head (present by default). This change did not affect the performance significantly.

### B.6 Learned embeddings for text-based number encodings

Figure 8 shows the structure of number embeddings learned on different datasets for different encodings. For P10 the models learn rotary structure which is reminiscent of other works such as grokking [27], and allows recovering relative numbers from inner products. It is also interesting to see how different datasets can lead to different learned encoding structures, for instance the arithmetic tasks seem to induce a more precise curve structure, while the planet data leads to more spread out embeddings, perhaps because the task is less sensitive to small perturbations of the numbers.

### B.7 Normalization via Layer-Norm

We verified empirically that Eq. equation 1 approximately holds. Figure 9 show such a curve.

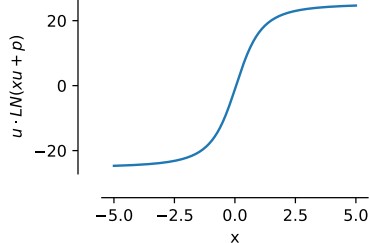

Figure 9: Value of the embedding of the number $x$ after layer-norm, projected onto the direction of the [NUM] embedding vector.

