# OpenReview forum: "xVal: A Continuous Number Encoding for Large Language Models"
_NeurIPS.cc/2023/Workshop/AI4Science — NeurIPS2023-AI4Science Poster_

### Official Review · Reviewer_wTP6 · 2023-10-20
**Interesting as an empirical study, but not a viable method yet**

**Rating:** 4
**Confidence:** 4

**Review:**

The manuscript introduces xVal, an embedding method for numeric values that simply multiplies that number with a single "`[NUM]`" token's learned vector embedding. Experiments use a synthetic dataset of arithmetic operations, a dataset of global temperature data, and a dataset of planetary orbit simulations. The method is tested in a Masked Language Model (MLM) setting, but could a priori be used in an Auto-Regressive (AR) setting.

I can appreciate the simplicity and directness of the method, but I have serious reservations concerning it's applicability in a realistic setting, and I find the general presentation of the manuscript misleading. If the same work were clearly presented as a report of empirical observations and/or preliminary results toward a method, then I could support accepting this as a poster in the workshop. But, unless I'm missing something major, not in it's current state.

Pros:
  - This is a simple approach that appears original to me, and it could inspire/inform future work.

Cons:
  - These preliminary results are misrepresented as a working method.
  - The actual envisioned use case isn't clearly described.
  - Some arguments do not hold outside the restrictive MLM context.

### Specific points follow.

Line 88:
> In the experiments of this paper, we normalize numbers in the text corpus such that they fall within the range [−5, 5] as a preprocessing step before training.

Such a normalization only makes sense for a specific context: you need to have some understanding of the ranges that the numbers may take before you can so renormalize them. This limitation precludes using the method for pre-training on a large, general corpus. The model may thus at best be used for fine tuning, and changing of number representation in fine-tuning would cause the model to discard a lot of what was learned in pre-training. Note that P10, P1000, B1999 and FP15 may all be used at pre-training.

This takes me to the issue that there is no clearly stated use case: how do the authors envision the use of their model in a real-world setting? The experiments arithmetic operations, global temperature and planetary orbits are all fine toy examples to understand the model's behavior. But a calculator can do arithmetic operations very well, and the two other datasets would be better handled by a tabular and/or time series model. **What real-world problem could be addressed with this method and yield advantages over alternatives?**

Line 159 mentions the JSON format not having a causal format as a justification for MLM, but I don't buy it: you have to parse the JSON to mask-out the last entry anyway, so you can reformat it anyway you choose for AR generation. In the worst case, you could use Fill-in-the-Middle (FIM).

The authors claim (e.g., lines 120-123) that alternative numerical encoding methods leverage spurious correlations. A specific argument relates to the number of tokens used to to encode a number, but note that this correlation may only be leveraged in a MLM setting, and most of the recent developments in LLMs involve AR generation.

More generally, there is an argument to be made that there is something to learn from a string representation of a number beyond it's magnitude. For example, 100 and 99.99 are very close in value to one another, but the latter implies a precision that the former didn't. My understanding is that some non-spurious correlations of that sort may be learned by other methods, and not by xVal.

### Random thoughts/comments:

Regarding the failure mode discussed on lines 247-261, here's how I understand it. In that example, the planet mass is hard to predict, and since xVal is trained by MSE (line 97), it has the easy "cop out" of just predicting the mean. In contrast, token-based numerical encodings have no benefit from predicting "near" tokens, so they must try to predict the correct one.

Relevant work regarding the arithmetic task: https://arxiv.org/pdf/2307.03381.pdf . TL;DR: Humans do arithmetic by figuring out digits in a right-to-left manner, because that's the direction that the algorithm makes sense. If all you have is a small transformer that decodes left-to-right, you should give it a little-endian notation, enabling 100% accuracy.

---

### Official Review · Reviewer_FMqM · 2023-10-25
**This work introduced XVAL, a continuous number encoding for LLM**

**Rating:** 7
**Confidence:** 4

**Review:**

The contribution is timely since many current LLMs struggle with numbers due to tokenization.

They contribute a novel Continuous Number Encoding scheme for LM and is proven to achieve good results both theoretically and experimentally.

As far as I know, LLaMa models encode each number individually and thus is very good at numerical operations. I am curious how your methods compare with theirs.

It is unclear how this method behaves on real 'large' LMs since the experiments are only conducted on small LMs.

Overall, I think the contribution is good to accept.

---

### Meta-Review · Area_Chair_eVAb · 2023-10-27

**Recommendation:** Accept (Poster)
**Confidence:** 3

**Metareview:**

The authors present a novel numerical encoding scheme called XVAL designed to represent any real number with a single token. This innovation aims to address tokenization challenges faced by LLMs when processing numerical data. While the methodology proposed is intriguing and exhibits potential for some applications, further clarification on its broader applicability and an expanded set of evaluations are necessary.

Strengths:

* Innovative Approach: The idea of using a continuous embedding space for numbers within LLMs is unique and addresses an existing gap in the literature. The scheme eliminates the multiple token issue faced by traditional tokenization processes, potentially making it more efficient.
* Clear Theoretical Presentation: The authors lay out the theory behind XVAL meticulously. The methods, including the number inference scheme and the implicit normalization via layer-norm, are detailed well.
* Potential in Scientific Domains: The proposed inductive bias for scientific applications is a compelling point of the study. If fully realized, this could open doors for more accurate analysis of scientific datasets with LLMs.

Weakness:

* Scope of Applicability: One strong limitation is the normalization process. Restricting the normalization to the range [−5, 5] is quite specific and might not be universally applicable. This raises concerns about XVAL's utility for pre-training on diverse datasets.

* Comparison with Other Models: A direct comparison with models like LLaMa, which have a track record of efficiently handling numerical operations, would have been beneficial. The absence of such comparisons leaves a gap in the evaluation.

* Real-world Applicability:  While the experiments mentioned are relevant, they seem more theoretical than applicable to real-world complex tasks.

* Value Interpretation and Precision: The method might not differentiate between numbers that are close in value but convey different precision. Such nuances are crucial in various applications, and how XVAL handles them remains unclear.